# Radiological Investigation on Sediments: A Case Study of Wadi Rod Elsayalla the Southeastern Desert of Egypt

**Ahmed E. Abdel Gawad** [1,†]**, Khaled Ali** [1]**, Hassan Eliwa** [2]**, M. I. Sayyed** [3,4]**, Mayeen Uddin Khandaker** [5]**, David A. Bradley** [6]**, Hamid Osman** [7]**, Basem H. Elesawy** [8] **and Mohamed Y. Hanfi** [1,9,*,†]

[1] Nuclear Materials Authority, Maadi, P.O. Box 530, Cairo 11936, Egypt; gawadnma@gmail.com (A.E.A.G.); Khaled_ali@yahoo.com (K.A.)
[2] Geology Department, Faculty of Science, Minufiya University, Shebin ElKom 32511, Egypt; eliwa98@yahoo.com
[3] Department of Nuclear Medicine Research, Institute for Research and Medical Consultations, Imam Abdulrahman Bin Faisal University, Dammam 31441, Saudi Arabia; mabualssayed@ut.edu.sa
[4] Department of Physics, Faculty of Science, Isra University, Amman 11622, Jordan
[5] Center for Applied Physics and Radiation Technologies, School of Engineering and Technology, Sunway University, Bandar Sunway 47500, Selangor, Malaysia; mayeenk@sunway.edu.my
[6] Department of Physics, University of Surrey, Guildford GU2 7XH, UK; d.a.bradley@surrey.ac.uk
[7] Department of Radiologial Sciences, College of Applied Medical Sciences, Taif University, Taif 21944, Saudi Arabia; ha.osman@tu.edu.sa
[8] Department of Pathology, College of Medicine, Taif University, P.O. Box 11099, Taif 21944, Saudi Arabia; b.elesawy@tu.edu.sa
[9] Institute of Physics and Technology, Ural Federal University, St. Mira, 19, 620002 Yekaterinburg, Russia
\* Correspondence: mokhamed.khanfi@urfu.ru or m.nuc2012@gmail.com
† Ahmed E. Abdel Gawad and Mohamed Y. Hanfi contributed equally to this work.

**Abstract:** The presence of heavy radioactive minerals in the studied granitoids from which the Wadi sediments leads to the study of the exposure to emitted gamma rays from the terrestrial radionuclides, such as $^{238}$U, $^{232}$Th, and $^{40}$K. The geological study revealed that the Wadi sediments derived from the surrounding granitoids, such as syenogranite, alkali feldspar granite, and quartz syenite. The mineral analysis confirmed that the granitoids were enriched with radioactive minerals, such as uranothorite as well as monazite, zircon, yttrocolumbite, and allanite. The mean activity of the $^{238}$U, $^{232}$Th, and $^{40}$K concentrations are 62.2 ± 20.8, 84.2 ± 23.3, and 949.4 ± 172.5 Bq kg$^{-1}$, respectively, for the investigated Wadi sediments, exceeding the reported limit of 33, 45 and 412 Bq kg$^{-1}$, respectively. Public exposure to emitted gamma radiation is detected by estimating many radiological hazard indices, such as the radium equivalent content (Ra$_{eq}$), external and internal hazard indices (H$_{ex}$ and H$_{in}$), annual effective dose (AED), annual gonadal dose equivalent (AGDE), and excess lifetime cancer (ELCR). The obtained results of the radiological hazards parameters showed that public exposure to emitted gamma radiation can induce various dangerous health effects. Thus, the application of the investigated sediments in different building materials and infrastructures fields is not safe. A multivariate statistical analysis (MSA) was applied to detect radionuclide correlations with the radiological hazard parameters estimated in the granite samples.

**Keywords:** Wadi sediments; heavy minerals; natural radionuclides; radiological hazard

## 1. Introduction

Natural radionuclides can be found in varying amounts in all rock types, depending on their concentration levels in the sources. Moreover, large levels of radionuclides have a harmful impact on public health and the surrounding environment. The principal gamma radiation sources in rocks, soils, and water are terrestrial radionuclides, such as $^{238}$U, $^{232}$Th, and the radioactive isotope $^{40}$K [1,2]. Gamma radiations from these radioisotopes are one of the most common external sources of population radiation [3,4]. Radiation

can make the human body vulnerable through a variety of processes, including external and internal exposures [5]. The enormous geochemical diversity of radionuclides in the atmosphere allows them to transit within a wide range of environments and infect much of the environment humans interact with. Natural radionuclides such as the $^{232}$Th and $^{238}$U chains, as well as $^{40}$K, have a half-life duration similar to that of the earth, although they still exist in all areas of the earth and are spread in varying quantities depending on geography and geological configuration [6–8]. In the most commonly mined ores, the $^{238}$U, $^{232}$Th, and $^{40}$K activity concentrations do not surpass the acceptable global averages of 33, 45, and 412, respectively [9,10]. The mentioned radionuclides contribute to the background levels. However, because of the large quantities of radioactive materials in urban soil, background levels of radiation are high all over the world. Granites are usefully utilized as building materials that are sources of radiation exposure, particularly if they contain a high proportion of natural or artificial radionuclides [11–15]. The sediments are derived from the surrounding granites due to weathering and aeration processes [16]. The sediments play an essential role in the surrounding areas; thus, the assessment of radioactive exposure is important for the people living around the investigated area. Based on the Agency for Toxic Substances and Disease Registry (ATSDR), the long period exposure to U, Ra, and their decay products causes serious illnesses, such as chronic lung disease, oral necrosis, leukopenia, and anemia [17,18]. In addition, liver, kidney, bone, lung, and pancreatic malignancies can all be induced by Th exposure [19]. Human activities, such as U mining, result in the release of radionuclides and their subsequent dispersion in the environment [20]. The sediments in the studied area can be utilized in various infrastructures applications. Therefore, the novelty of the present work is evaluating the presence of the radionuclides $^{238}$U, $^{232}$Th, and $^{40}$K and their activity concentrations in the collected sediments samples from the studied area (Wadi Rod Elsayalla area, Southeastern Desert of Egypt, Figure 1). The assessment of radiological hazards was detected with various radioactive parameters, such as radium equivalent activity ($Ra_{eq}$), the absorbed dose rate ($D_{air}$), the annual effective dose (AED), external ($H_{ex}$), and internal hazard ($H_{in}$) indices. Moreover, the annual gonadal dose equivalent (AGDE) and the excess lifetime cancer risk (ELCR) were computed.

## 2. Geologic Setting

The Wadi Rod Elsayalla area is located in the Southeastern Desert of Egypt. It can be reached through asphaltic rod about 120 km eastward of Aswan city and occurs between latitude 23°50′47″ and 23°53′28″ N and longitude 34°20′52″ and 34°23′44″ E (Figure 2a,b).

The exposed rock units of Wadi Rod Elsayalla are represented by metavolcanics, syenogranite, alkali feldspar granite, and quartz syenite dissected by microgranite dikes as well as quartz veins (Figure 1a–f). Several studies have been carried out on the area under investigation by many authors [21,22].

Metavolcanic rocks form a thick sequence of stratified lava flows interbedded with their pyroclastics and intruded by granitoid (Figure 2, [22]). They are composed mainly of thick lava flows of dark green to grey with some metabasalt, metadacite, and metarhyolite thrown in for good measure. Metatuffs are very fine grained and greyish green, light to dark green, black, and buff in color. They are banded, jointed, exfoliated, weathered, and highly altered. They are ash and lithic lapilli tuffs (Figure 1a). These rocks are basaltic, andesitic, and dacitic to rhyolitic in compositions. Metabasaltic rocks are composed essentially of plagioclase laths and pyroxene set in a fine-grained groundmass. Meta-andesitic rocks, which are of the porphyritic type, are composed of plagioclase hornblende, biotite, quartz, and iron oxides. Metadacite rocks are composed mainly of plagioclase, K-feldspar, quartz, and fine biotite flakes set in a very fine-grained groundmass. Metarhyolite rocks are composed of quartz, K-feldspars, and plagioclase phenocrysts embedded in a fine groundmass. The metavolcanics have an age dated between 860 and 825 Ma [23].

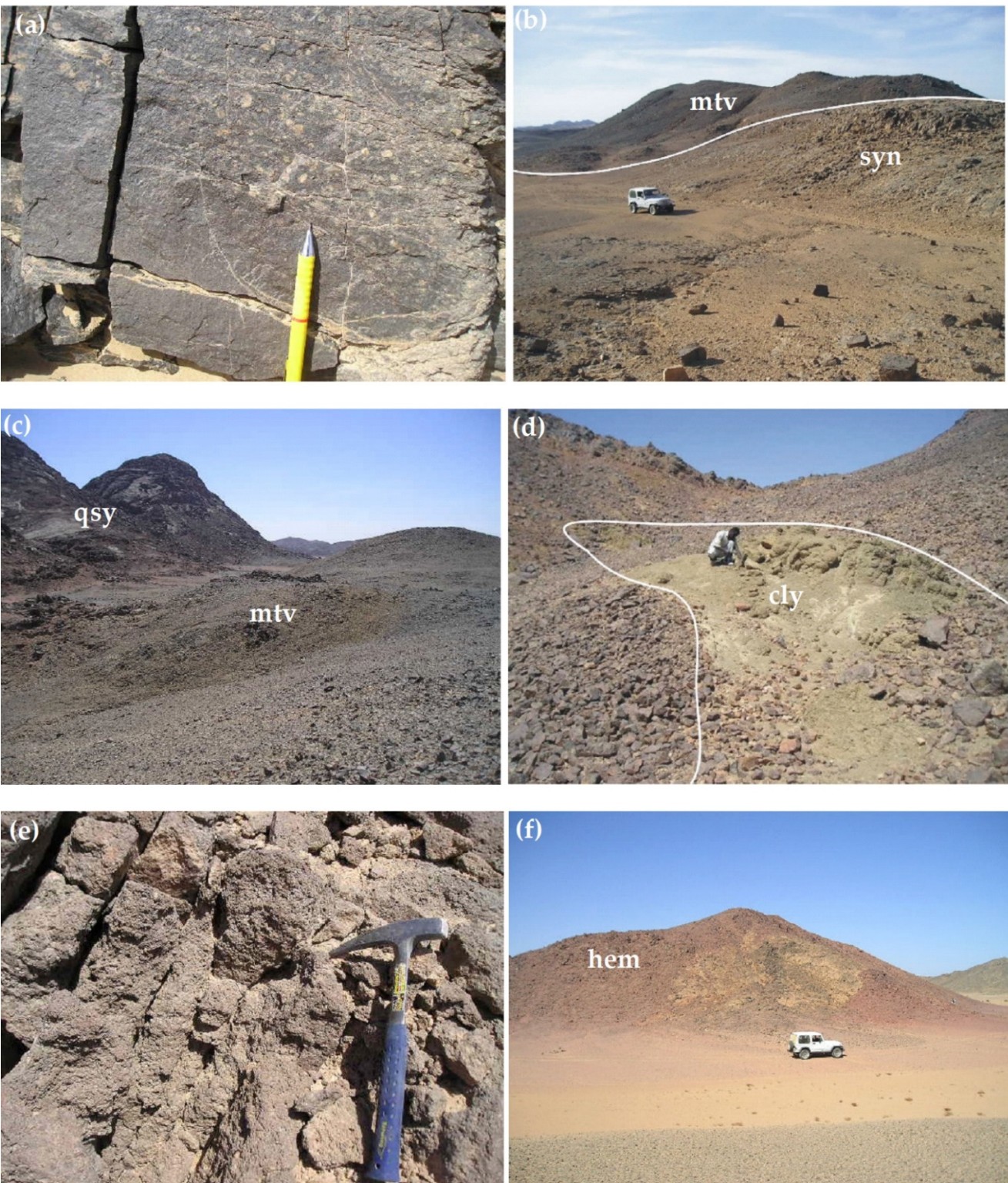

**Figure 1.** (**a**) Lapilli metatuffs associating light lithic fragments, (**b**) Syenogranite intrudes metavolcanics (mtv) with sharp contact, (**c**) Quartz syenite (qsy) intrudes metavolcanics (mtv), (**d**) Quartz syenite shows intensive clay alteration (cly), (**e**) Syenogranite exposed to episyenitization processes (quartz dissolution), (**f**) Syenogranite highly altered to hematitization (hem).

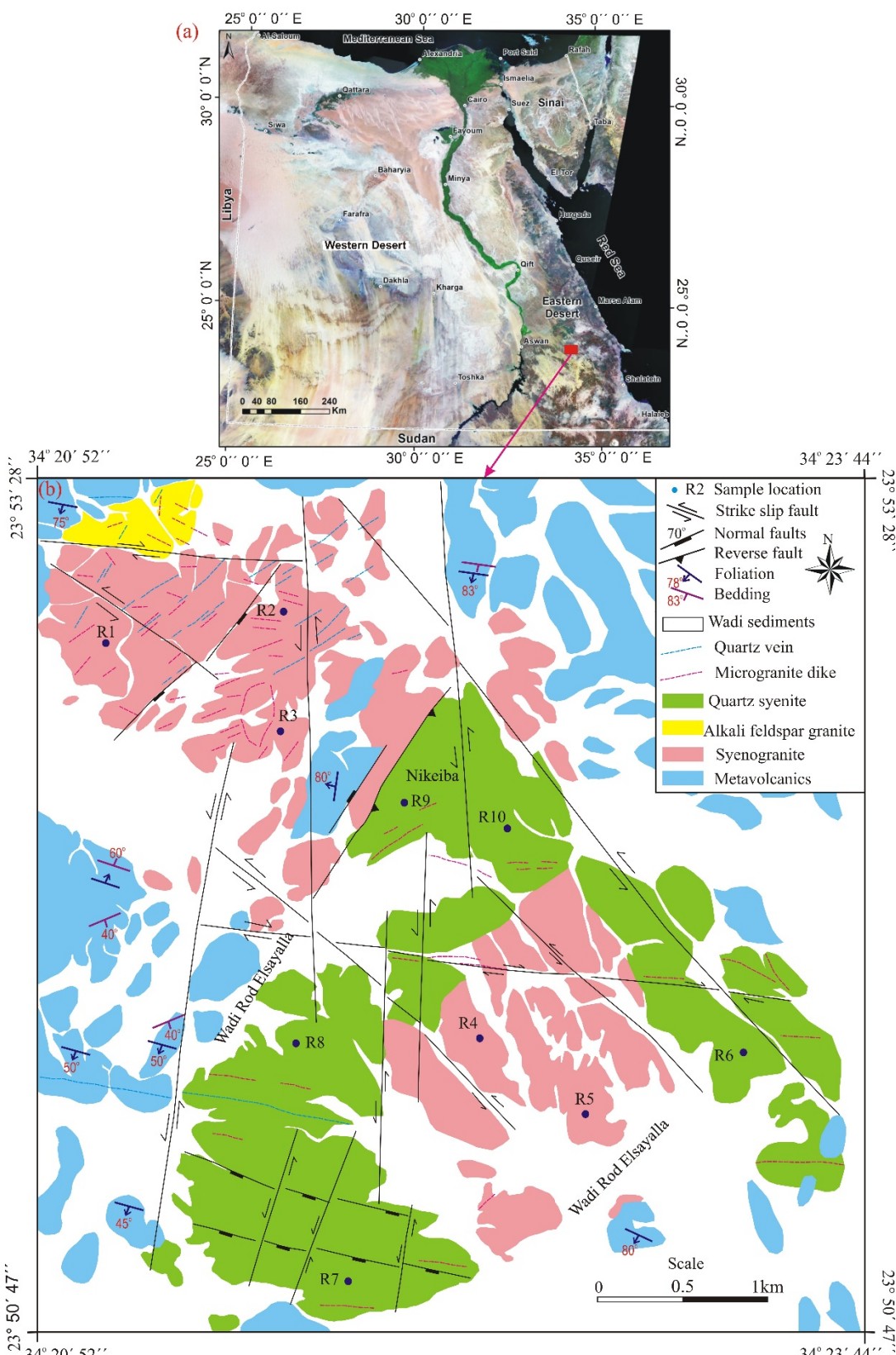

**Figure 2.** (**a**) Physiographic map and (**b**) Detailed geologic map of Wadi Rod Elsayalla, Southeastern Desert, Egypt according to [22].

Syenogranite is medium to coarse grained and reddish to pale pink in color. It is highly weathered, jointed, strongly altered, and exfoliated. Syenogranite is composed principally

of K-feldspar, quartz, biotite, plagioclase, and iron oxides. They contain xenoliths up to 1m of subangular metatuffs along their outer periphery. Alkali feldspar granite is coarse grained, whitish to pale white in color, and highly weathered. Quartz syenite is medium to coarse grained and dark grey to pale greenish grey or pale pink in color and has a moderate to high relief. It is highly weathered and exfoliated and composed of K-feldspars, quartz, biotite, riebeckite, arfvedsonite, and a very subordinate amount of plagioclase. Greanitoids intrude metavolcanics with sharp contact (Figure 1b,c), and they are highly altered into clay alteration, episyenitization, and hematitization (Figure 1d–f).

The granitoids rocks in the Eastern Desert of Egypt vary in their composition from syenogranite and monzogranite to alkali feldspar granite, and they are dated between 610 and 550 Ma during the Late-African Orogeny [24–27].

Pegmatite pockets are unmappable rocks and occur as irregular and small bodies in syenogranite rocks. They are very coarse grained, red to buff in color, and mainly composed of K-feldspar, quartz, biotite, plagioclase, and muscovite. The younger granite intrusions and their related pegmatites are dated $610 \pm 20$ and $594 \pm 12$ Ma [28].

Microgranite dikes crosscut the metavolcanis and granitoid of the Wadi Rod Elsayalla area. They range from 1 to 2 m in width and are pale pink, fine grained, and composed mainly of K-feldspar, quartz, plagioclase, and biotite. They strike NE-SW and E-W with nearly vertical or steeply dipping, straight or slightly curving. These dikes are highly altered and enclose manganese oxides as thin films along fracture planes. Quartz veins and veinlets predominate in the studied metavolcanics and granitoid. They are milky white or green to blue in color due to the presence of amazonite minerals in the syenogranite and are sometimes stained red by iron oxides. These veins vary in thickness from 10 cm to 20 m. They strike NE-SW, E-W, and NW-SE and are steeply dipping to nearly vertical.

The Wadi Rod Elsayalla are composed essentially of recent sediments covered mainly by a dry drainage network which is identified as Wadis (streams). These Wadis are linear and could be formed as a result of structural control. They show variations in their width and length and are filled with angular, subangular to subrounded rock fragments of the basement rocks, including metavolcanics, syenogranite, alkali feldspar granite, quartz syenite, and microgranite as well as quartz. The rock fragments range from large boulders to sand in size and are mixed with silt and clay particles. The soil clays are abundant in the studied area due to the higher slope of the erosion zone associated with vegetation.

## 3. Materials and Methods

### 3.1. GS-256 Spectrometer

A Geophysica Brno GS-256 Gamma Spectrometer with a 0.35 L sodium iodide (NaI) thallium-activated detector performed the ground gamma-ray spectrometric observations on Wadi Rod Elsayalla. A measuring interval of 120 s was adjusted to allow sufficient time in order to establish a stable spectrum for reading radioelements. The survey sites were selected to cover most of the exposures of Wadi Rod Elsayalla. The recorded radioelement was repeated at least three times for good statistics in each site for eTh (ppm) and eU (ppm) and K (%) at each measuring station. The GS integrates a horizontal area of about 1 m diameter with nearly 25 cm for depth when in direct contact with the outcrops. Before the field survey, the calibration of the spectrometer was established using synthetic concrete pads at Egypt's Nuclear Materials Authority. The pads contained known concentrations of potassium, uranium, and thorium as described in [29].

### 3.2. Heavy Minerals

Ten mineralized samples were collected from the highest radioactive zones from syenogranite (5) and quartz syenite (5) in order to investigate the heavy minerals, especially the radioactive ones (Figure 2b). These samples were subjected to disintegration (crushing and grinding). The samples were washed using water. The slimes were removed, and then the samples were dried. The dried fraction was exposed to be sieved into the size −0.5 to +0.063 mm. The systematic mineral separation technique, including a water table

and heavy liquid (Bromoform, 2.8 g/cm$^3$ for specific gravity), was used to separate the light and heavy fractions. In addition, polished sections were selected from granitoids for mineralogical investigation.

### 3.3. Methods of Separation

The Isodynamic Frantz Magnetic Separator was carried out on the obtained mineral fractions. The heavy minerals were separately picked as individual grains using the Stereo Microscope. Then we used an environmental scanning electron (ESEM) provided with an energy dispersive spectrometer (EDS) unit (model Philips XL 30 ESEM) at the Nuclear Materials Authority in Cairo, Egypt. The diameter of the analyzed area was (1–2 mm) at 25–30 kV accelerating voltages, with counting times varying between 60 and 120 s with detection limits varying between 0.1 and 1 wt%. The quantitative result had an accuracy of 2–10 wt% for elements Z > 9 (F) and 10–20 wt% for the light elements B, C, N, O, and F, which were employed to identify the separated heavy minerals, particularly the radioactive ones.

## 4. Results and Discussion

### 4.1. Mineralogical Features

The sediments associated with the surrounding granitoids are enriched with several radioactive minerals. Thus, the investigated mineralization associated with syenogranite is uranothorite, monazite, zircon, and yttrocolumbite.

*Uranothorite* (Th, U) SiO$_4$ is considered the prime carrier of Th and U in syenogranite. The EDS analysis shows uranothorite is mainly composed of Th (56.21 wt%), U (15.85 wt%), and Si (10.75 wt%). It contains some rare earth elements (REE), especially Y (2.75 wt%) and Yb (2.17 wt%) and is interstitially stained by iron oxides (Figure 3a).

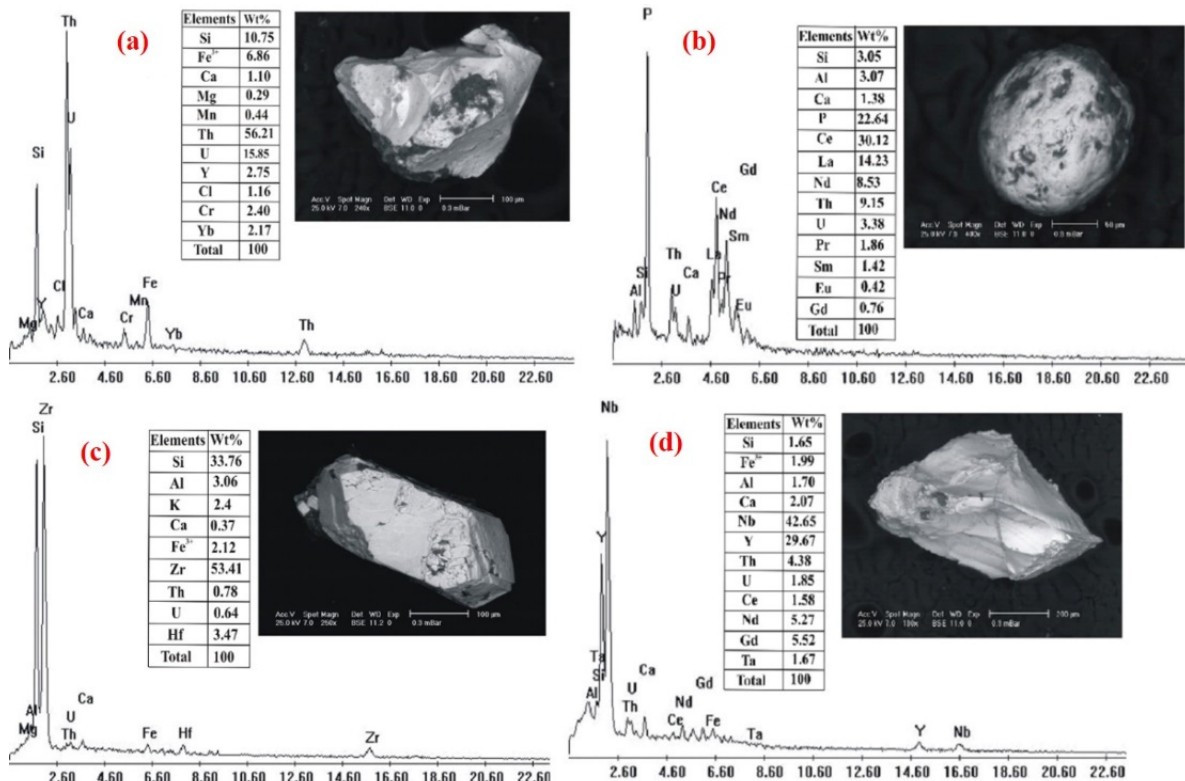

**Figure 3.** BSE images and EDS analyses for (**a**) uranothorite (**b**) monazite (**c**) zircon (**d**) yttrocolumbite from syenogranite rocks, Wadi Rod Elsayalla, Southeastern Desert, Egypt.

*Monazite*-(Ce) (Ce, La, Pr, Nd, Th, and Y) PO$_4$ is a phosphatic REE mineral containing and is considered a good source of Th, La, and Ce. It occurs as a subrounded to rounded

oval, and sometimes has tabular transparent isolated grains, and it is medium to coarse grained. The pits and grooves are common features on the surface of monazite grains. The chemical composition of monazite is revealed in (Figure 3b). The EDS analysis of monazite shows that the contents of P (22.64 wt%) and rare earth elements (REE) are La, Ce, Pr, Nd, Sm, Eu, and Gd (14.23, 30.12, 1.86, 8.53, 1.42, 0.42, and 0.76 wt%, respectively). Some monazite grains contain high Th (9.15 wt%) with relatively low U (3.83 wt%). Thorium is a highly radioactive element in monazite and could substitute for U and LREE. $ThO_2$ could reach up to 8.25 wt% and 11.69 wt%, and the $UO_2$ contents reached are 0.57 and 4.83 wt% in monazite [30,31]. The Ce has a marked predominance over the other REEs designating monazite-(Ce).

*Zircon* ($ZrSiO_4$) is found in the form of euhedral and prismatic crystals with opaque microinclusions. It varies from brown to grey to red in color with variable sizes. The EDS analysis of zircon crystals from syenogranite is mainly composed of Zr (53.41 wt%), Si (33.76 wt%), and Hf (3.47 wt%). Th and U are recorded in low contents. Iron reached 2.12 wt% in the analyzed zircon grain (Figure 3c).

*Yttrocolumbite* mineral is one of columbite group that comprises a general formula $AB_2O_6$, A = Fe, Mn Y, REE, Ca, U, and Th, and B = Nb and Ta. It is a synonym for samarskite. It is composed mainly of complex oxides of Nb, Ta, and Y-group and rare earth elements with significant contents of both Th and U as well as Ca and $Fe^{3+}$ (Figure 3d). The EDS analysis shows that yttrocolumbite is essentially composed of Nb (42.65 wt%) and Y (29.67 wt%) with a minor content of Ta (1.67 wt%). Yttrocolumbite carries potential contents of REE especially Ce, Nd, and Gd (1.58, 5.27, and 5.52 wt%, respectively), and Th reaches (4.38 wt%), and U reaches (1.85 wt%).

On the other hand, quartz syenite rocks are enriched with radioactive bearing minerals, such as zircon and allanite.

*Zircon:* ($ZrSiO_4$) occurs as euhedral prismatic as microinclusions in allanite crystals (Figure 4). Others are anhedral crystals and contain microinclusions of opaque minerals and occur along the peripheries of allanite crystals. The general reddish-brown color may be due to a content of iron (1.97 to 9.57 wt%) (Table 2). Zircon shows a marked enrichment in Hf, U, and Th, especially Zr2 and Zr5 (Table 2, Figure 4). The EDS analyses of zircon grains (Table 2) show that they are composed essentially of Zr ranging from 48.68 to 56.78 wt% and Si between 24.27 and 35.23 wt%. Other elements are recorded as minor amounts such as Al, K, and Ca. Zircon occurs in the early-formed biotite flakes as well as in potash feldspars, suggesting that the examined zircon and its hosting granites originated from water-rich magma.

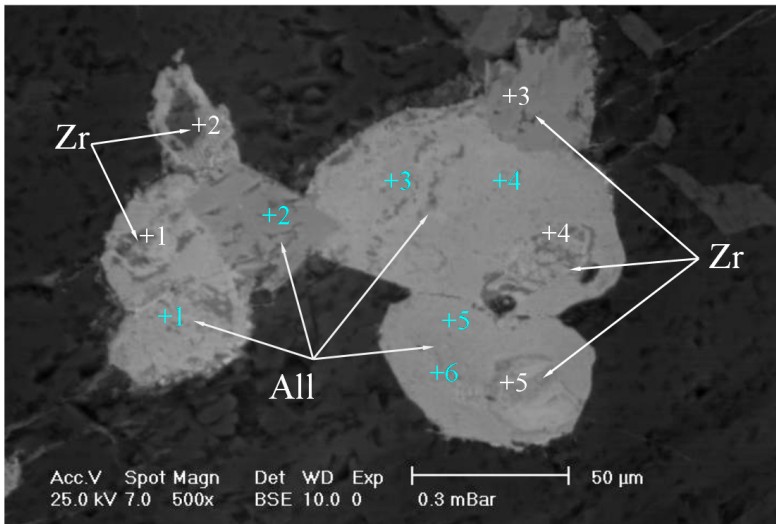

**Figure 4.** BSE images for allanite crystals enclosed zircon crystals from quartz syenite, Wadi Rod Elsayalla, Southeastern Desert, Egypt as mentioned in (Tables 1 and 2).

**Table 1.** The semiquantitative (elements, wt%) results of EDS analyses for allanite crystals from quartz syenite, Wadi Rod Elsayalla, Southeastern Desert, Egypt.

| Elements | A1 | A2 | A3 | A4 | A5 | A6 |
|---|---|---|---|---|---|---|
| Si | 16.79 | 23.23 | 12.94 | 18.27 | 14.55 | 10.15 |
| Al | 4.64 | 13.04 | 6.12 | 9.39 | 4.95 | 3.03 |
| Ca | 3.32 | 3.2 | 6.5 | 3.25 | 3.24 | 3.98 |
| $Fe^{3+}$ | 3.67 | 7.41 | 3.27 | 5.73 | 4.06 | 3.31 |
| K | 0.81 | 2.1 | 1.41 | 3.01 | 1.14 | 0.98 |
| Na | n.d. | n.d. | n.d. | n.d. | 1.99 | 1.79 |
| Ti | n.d. | n.d. | n.d. | n.d. | 11.7 | 14.6 |
| Ce | 32.16 | 21.62 | 33.32 | 26.09 | 22.6 | 24.61 |
| La | 18.73 | 9.35 | 18.67 | 15.28 | 12.91 | 12.62 |
| Nd | 8.29 | 7.41 | 7.06 | 6.76 | 7.66 | 7.23 |
| Pr | 2.28 | 2.51 | 2.26 | 2.25 | 1.30 | 2.40 |
| Y | 4.20 | 4.03 | 4.69 | 6.47 | 4.99 | 5.0 |
| Th | 2.18 | 2.78 | 0.64 | 1.67 | 2.51 | 2.52 |
| U | 0.81 | 1.09 | 0.46 | 0.86 | 1.16 | 1.13 |
| S | 0.34 | 0.29 | 0.39 | 0.48 | n.d. | n.d. |
| Sm | 0.24 | 0.51 | n.d. | n.d. | 1.52 | 1.36 |
| Eu | 0.26 | 0.31 | 0.56 | n.d. | 0.27 | 0.32 |
| Gd | 1.27 | 1.12 | 1.71 | 0.51 | 0.82 | 1.17 |
| Nb | n.d. | n.d. | n.d. | n.d. | 2.63 | 3.8 |
| Total | | | | 100 | | |

n.d. (not determined).

**Table 2.** The semiquantitative results (elements, wt%) of EDS analyses of zircon crystals from quartz syenite, Wadi Rod Elsayalla, Southeastern Desert, Egypt.

| Elements | Zr1 | Zr2 | Zr3 | Zr4 | Zr5 |
|---|---|---|---|---|---|
| Si | 28.04 | 24.72 | 32.75 | 35.23 | 24.93 |
| Al | 3.04 | 1.37 | 3.49 | 1.92 | 1.94 |
| K | 2.04 | 0.69 | 1.65 | 1.17 | n.d. |
| Ca | 0.66 | 1.18 | 1.23 | 0.29 | 1.34 |
| $Fe^{3+}$ | 6.71 | 9.57 | 4.55 | 4.54 | 1.97 |
| Zr | 50.2 | 54.62 | 48.68 | 53.01 | 56.78 |
| Th | 1.0 | 1.06 | 1.30 | 0.4 | 1.90 |
| U | 1.49 | 0.65 | n.d. | n.d. | 1.85 |
| Hf | n.d. | 6.14 | 1.8 | 3.44 | 8.54 |
| Y | 6.82 | n.d. | n.d. | n.d. | n.d. |
| La | n.d. | n.d. | 1.23 | n.d. | n.d. |
| Ce | n.d. | n.d. | 2.18 | n.d. | n.d. |
| Nd | n.d. | n.d. | 1.15 | n.d. | n.d. |
| Total | | | 100 | | |

n.d. (not determined).

*Allanite-(Ce)* (Ca; Ce)$_2$(Al; Fe$^{2+}$; Fe$^{3+}$)$_3$(SiO$_4$)(Si$_2$O$_7$)O(OH) is one of the main light rare earth minerals (LREE) and occurs as zoned euhedral to subhedral tabular prismatic crystals surrounded by zircon and sometimes enclosed zircon grains (Figure 4). It is usually intimately associated with biotite. The EDS data show that allanite is composed essentially of Ce ranging from 21.62 to 33.32 wt%, La 9.35 to 18.73 wt%, Nd 6.76 to 8.29 wt%, and Pr 1.3 to 2.4 wt%, while Y ranges from 4.03 to 6.47 wt%. Si ranges from 10.15 to 23.23 wt%, and Al is between 3.03 and 13.04 wt%. It is mostly coated by iron oxides, and hence Fe ranges from 3.31 to 7.41 wt%. Other elements occur as minor constituents, such as Ca, K, Na, Ti, S, Sm, Eu, Gd, and Nb (Table 1). So, allanite is enriched in both Ce and Fe designating ferriallanite-(Ce) in its chemical composition.

It is clarified that the studied area is important for extracting the radioactive material from the surrounding granites in the Wadi Rod Elsayalla area [11].

*4.2. Radionuclide Activity Concentrations*

The measurement data of the $^{238}$U, $^{232}$Th, and $^{40}$K activity concentrations of 380 Wadi sediments are presented in Table S1 (Supplementary Materials). The spectrometric measurements were performed directly in the field. Table 3 displays the descriptive statistical analysis (mean, standard deviation, minimum, maximum, skewness, kurtosis, and coefficient of variance) for the radionuclide activity concentrations besides the radiological hazard parameters. The mean values Mean ± SD (Min/Max) of the $^{238}$U, $^{232}$Th, and $^{40}$K activity concentrations are registered as 62.2 ± 20.8 (13.6/130.9), 84.2 ± 23.3 (30.0/192.0), and 949.4 ± 172.5 (375.6/1471.1) Bq kg$^{-1}$, respectively. The mentioned mean values of the $^{238}$U, $^{232}$Th, and $^{40}$K activity concentrations are approximately two times higher than the recommended values 33, 45, and 412 Bq kg$^{-1}$, respectively [1]. As can be noticed in Table 3, the activity concentration of $^{232}$Th in the studied Wadi sediments is greater than $^{238}$U, and thorium is less susceptible to leaching than uranium, thus the increase in thorium levels is usually larger due to the resistance of thorium minerals to weathering [32]. In addition, the radionuclide concentration elevations in the Wadi sediments can be depicted due to the weathering, leaching, alteration, and aeration processes. This leads to the accumulation of a high content of radioactive minerals, such as uranothorite as well as radioactive bearing minerals, such as monazite, zircon, allanite, and yettrocolombite, and the different alteration zones (episyenitization, kaolinization, albitization, and hematitization) from the surrounding granitoids, such as synogranite, alkali feldspar granite, and quartz syenite. Due to the presence of a wide range of minerals in potassium, such as feldspar, micas, and clay, the measured results depict that the $^{40}$K activity in the analyzed samples is greater than that of $^{238}$U and $^{232}$Th. The skewness values describe the asymmetric distribution based on the fundamental statistical analysis of radionuclide activity concentrations, with positive values referring to the asymmetrical distribution. Whereas their negative values suggest that the tail of the asymmetrical distribution is extended to negative values, the tail of the asymmetrical distribution is extended to negative values. Therefore, the positive skewness values of the $^{238}$U and $^{232}$Th activity concentrations imply a positive asymmetrical nature, whereas the negative skewness values of the $^{40}$K activity concentrations indicate a negative asymmetrical nature. Secondly, the kurtosis values reflect the Preakness probability distribution. The probability distribution is peaked in the analyzed study because the kurtosis values for the $^{238}$U, $^{232}$Th, and $^{40}$K activity concentrations are positive. As clarified in Table 3, the standard deviation data of all investigated samples are observed to be small and lower than the mean of the $^{238}$U, $^{232}$Th, and $^{40}$K activity concentrations, which reveals a high degree of uniformity of the detected radionuclides in the Wadi sediments. The coefficient of variance (CV) is presented in Table 3 with moderate values of 33% and 28% for the $^{238}$U and $^{232}$Th activity concentrations, respectively, while the low value was 18% $^{40}$K in the analyzed data. The variation may be due to the existence of the host minerals of uranium and thorium in the examined samples.

**Table 3.** Natural radionuclides and radiological hazard of Wadi sediments in Wadi Rod Elsayalla, Egypt.

| | N | Mean | SD | Min | Max | Lower 95% CI of Mean | Upper 95% CI of Mean | Variance | Skewness | Kurtosis | CV |
|---|---|---|---|---|---|---|---|---|---|---|---|
| U-238 | 380 | 62.2 | 20.8 | 13.6 | 130.9 | 60.1 | 64.3 | 433 | 0.70 | 0.60 | 0.33 |
| Th-232 | 380 | 84.2 | 23.3 | 30.0 | 192.0 | 81.9 | 86.6 | 541 | 1.20 | 2.91 | 0.28 |
| K-40 | 380 | 949.4 | 172.5 | 375.6 | 1471.1 | 932.0 | 966.8 | 29,753 | −0.11 | 0.31 | 0.18 |
| Ra$_{eq}$ | 380 | 255.7 | 50.6 | 107.6 | 433.8 | 250.6 | 260.8 | 2563 | 0.70 | 1.10 | 0.20 |
| H$_{in}$ | 380 | 0.9 | 0.2 | 0.4 | 1.5 | 0.84 | 0.88 | 0.033 | 0.72 | 0.83 | 0.21 |
| H$_{ex}$ | 380 | 0.7 | 0.1 | 0.3 | 1.2 | 0.68 | 0.70 | 0.019 | 0.70 | 1.10 | 0.20 |
| I$\gamma$ | 380 | 0.9 | 0.2 | 0.4 | 1.6 | 0.93 | 0.96 | 0.033 | 0.64 | 1.04 | 0.19 |
| Dair | 380 | 118.5 | 22.8 | 50.2 | 197.3 | 116.2 | 120.8 | 518 | 0.64 | 1.00 | 0.19 |
| AEDout | 380 | 0.1 | 0.0 | 0.1 | 0.2 | 0.1425 | 0.15 | 0.001 | 0.64 | 1.00 | 0.19 |
| AEDin | 380 | 0.6 | 0.1 | 0.2 | 1.0 | 0.5701 | 0.59 | 0.012 | 0.64 | 1.00 | 0.19 |
| AGDE | 380 | 0.8 | 0.2 | 0.4 | 1.4 | 0.8 | 0.86 | 0.025 | 0.61 | 0.97 | 0.19 |
| ELCR | 380 | 0.0005 | $9.8 \times 10^{-5}$ | $2.2 \times 10^{-4}$ | $8.5 \times 10^{-4}$ | 0.0005 | 0.0005 | $9.54 \times 10^{-9}$ | 0.64 | 1.00 | 0.19 |

SD = Standard deviation, Min = Minimum, Max = Maximum, CV = Coefficient of Variance, N = Number of samples.

The comparison between the present results of the $^{238}$U, $^{232}$Th, and $^{40}$K activity concentrations with the previous review data is listed in Table 4. This comparison shows the radionuclide activity concentrations results pivot essentially on the geological description of any studied area. The present results of the $^{238}$U, $^{232}$Th, and $^{40}$K activity concentrations display either lower or similar to the previous data.

**Table 4.** Comparison of $^{238}$U, $^{232}$Th and $^{40}$K activity concentration in Wadi Rod Elsayalla area with numerous studies from around the world.

| Country | $^{238}$U | $^{232}$Th | $^{40}$K | References |
|---------|-----------|------------|----------|------------|
| Egypt | 62 | 84 | 949 | Present study |
| Egypt | 137 | 82 | 1082 | [33] |
| Nigeria | 63.29 | 226.67 | 832.59 | [34] |
| Saudi Arabia | 28.82 | 34.83 | 665.08 | [35] |
| Saudi Arabia | 11 | 22 | 641 | [36] |
| Palestine | 71 | 82 | 780 | [37] |
| Jordan | 41.52 | 58.42 | 897 | [38] |
| Oman | 17 | 18 | 379 | [39] |
| Iran | 77.4 | 44.5 | 1017.2 | [40] |
| Malaysia | 184 | 165 | 835 | [41] |
| India | 25.88 | 42.82 | 560.6 | [42] |
| China | 112 | 72 | 672 | [43] |
| Spain | 84 | 42 | 1138 | [44] |
| Greece | 74 | 85 | 881 | [45] |

### 4.3. Radiological Hazards Parameters

Exposure of the public to radiation of the Wadi sediments can be estimated via several radiological hazards parameters, such as radium equivalent activity ($Ra_{eq}$), internal and external hazard indices ($H_{in}$ and $H_{ex}$), absorbed dose rate ($D_{air}$), annual effective dose (AED) in indoor and outdoor, annual gonads dose equivalent (AGDE), and excess lifetime cancer risk (ELCR).

### 4.3.1. Radium Equivalent Activity ($Ra_{eq}$)

The radium equivalent content ($Ra_{eq}$) can be used to detect the nonuniform distribution of $^{238}$U, $^{232}$Th, and $^{40}$K in the Wadi sediments. The values of $Ra_{eq}$ must be smaller than 370 Bq kg$^{-1}$, which identify the same annual dose of 10, 7, and 130 Bq kg$^{-1}$ for $^{238}$U, $^{232}$Th, and $^{40}$K, respectively. The data of $Ra_{eq}$ can be computed by the following Equation (1) [2,46]:

$$Ra_{eq} = A_U + 1.43\, A_{Th} + 0.077\, A_K \tag{1}$$

where $A_U$, $A_{Th}$, and $A_K$ refer to the activity concentration of $^{238}$U, $^{232}$Th, and $^{40}$K, respectively. Based on Equation (1), the $Ra_{eq}$ mean value (255.7 Bq kg$^{-1}$) of the investigated Wadi sediments was observed to be less than the approved level (370 Bq kg$^{-1}$) for public dose. However, the maximum value (433.8 Bq kg$^{-1}$) was found to be greater than the approved level, which indicates the derived Wadi sediments of the surrounding rocks contain high concentrations of uranium and thorium, while the minimum value 107.6 Bq kg$^{-1}$was detected in the investigated Wadi sediments.

### 4.3.2. External and Internal Hazard Indices ($H_{in}$ and $H_{ex}$)

The significant health effect can be predicted by the $H_{ex}$ and $H_{in}$ hazard indices. The guidance safety level of $H_{ex}$ and $H_{in}$ data must be smaller than unity [47,48]. According to the $^{238}$U, $^{232}$Th, and $^{40}$K activity concentrations in the investigated Wadi sediments, the values of $H_{ex}$ and $H_{in}$ are computed using the relations (2) and (3) [47,48] and listed in Table 3:

$$H_{ex} = \frac{A_U}{370} + \frac{A_{Th}}{259} + \frac{A_K}{4810} \leq 1 \tag{2}$$

$$H_{in} = \frac{A_U}{185} + \frac{A_{Th}}{259} + \frac{A_K}{4810} \leq 1 \tag{3}$$

Table 3 reveals the data of $H_{ex}$ altered from 0.3 to 1.2 with a mean value of 0.7, which is less than the approved level ($H_{ex} < 1$). While the mean value of $H_{in}$ (0.9) was comparable with the approved level ($H_{in} < 1$), the minimum and maximum values are 0.4 and 1.5, respectively. Figure 5 illustrates the high values of $H_{ex}$ and $H_{in}$ in the studied Wadi sediments, which were 3.2% and 18.9%, respectively, from all their registered values. This means the Wadi sediments in the studied area may contribute significantly to health effects associated with gamma radiation, radon gas, and its daughters [49].

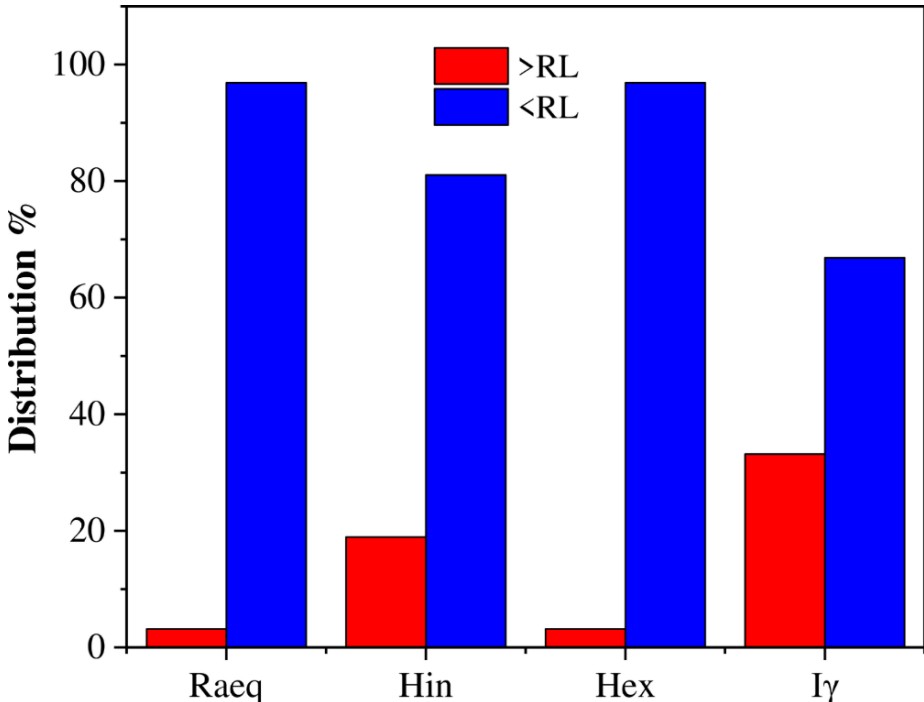

**Figure 5.** The distribution (%) of calculated radium equivalent content, internal hazard index, external hazard index, and gamma level index compared to the reported limit.

### 4.3.3. Representative Gamma Level Index ($I\gamma$)

The $I_\gamma$ index applied to identify the safe level of the investigated materials includes the radionuclide. The following relation (4) was utilized to compute $I_\gamma$ values [50]:

$$I_\gamma = \frac{A_U}{150} + \frac{A_{Th}}{100} + \frac{A_K}{1500} \leq 1 \tag{4}$$

The values of $I_\gamma$ varied from 0.4 to 1.6 with a mean value of 0.9. Figure 5 shows that the $I_\gamma$ high values represent 33% of the investigated Wadi sediments and also 5 illustrates that the studied sediments pose more health risk and are not candidates to utilize in building materials.

### 4.3.4. Absorbed Dose Rate ($D_{air}$) and Annual Effective Dose (AED)

The exposure to the emitted gamma rays from uranium, thorium, and potassium over 1 m from the surface can be described by the absorbed dose ($D_{air}$). The following Formula (5) was used to calculate the values of $D_{air}$ [51], which are presented in Table 3:

$$D_{air} = 0.430\, A_U + 0.666\, A_{Th} + 0.042\, A_K \tag{5}$$

The $D_{air}$ values of the Wadi sediments varied between 50.2 and 197.3 nGy h$^{-1}$ with a mean value of 1185 nGy h$^{-1}$ that was greater than the worldwide average of 59 nGy h$^{-1}$ [1].

Consequently, restrictions must be applied to prevent the use of Wadi sediments materials in the infrastructure fields. Therefore, the public's exposure to emitted gamma radiation is assessed by two exposure scenarios, both indoors and outdoors. The public spends 20% and 80% of their time outdoors and indoors, respectively. The AED was detected utilizing Equation (6) and is listed in Table 3 [16,52]:

$$\text{AED (mSv/y)} = D_{air} \times F \times T \times DCF \times 10^{-6} \tag{6}$$

F represents the occupancy factor, T is the exposure time to gamma radiation (8760 h), and DCF represents the dose conversion factor (0.7 Sv/Gy). The corresponding statistical values of the annual outdoor effective dose (AED$_{out}$) and annual outdoor effective dose (AED$_{in}$) are presented in Table 3. The mean values of AED$_{out}$ and AED$_{in}$ are 0.1 and 0.6 mSv y$^{-1}$, respectively and are 1.5 times higher than the recommended limits of 0.07 and 0.41 mSv y$^{-1}$ [1]. The presence of uranium and thorium minerals such as uraninite, thorianite, and monazite can result in exposure to high doses that cause adverse health impacts, such as the deterioration of tissues and deoxyribonucleic acid (DNA) in the genes, as well as cancer and cardiovascular disease [53].

### 4.3.5. Annual Gonadal Dose Equivalent (AGDE)

The AGDE is the radiological parameter employed to estimate the received dose received by organs yearly for humans, particularly the gonads. The values of the AGDE depend on the emitted gamma from the terrestrial radionuclides and are estimated according to the following Equation (7) [47]:

$$\text{AGDE (mSv y}^{-1}) = 3.09A_U + 4.18A_{Th} + 0.314A_K \tag{7}$$

Table 3 displays the statistical AGDE data computed for all data of the Wadi sediments in the studied area. The mean value of the AGDE was $0.8 \pm 0.2$ mSv y$^{-1}$ and was less than the approved limit of 0.3 mSv y$^{-1}$ [1]. The values of the AGDE ranged between 0.4 and 1.4 mSv y$^{-1}$. The highest value of the AGDE means that the Wadi sediments are not safe to apply in public activities, such as building materials, etc.

### 4.3.6. Excess Lifetime Cancer Risk (ELCR)

The public's exposure to gamma radiation, either outdoors or indoors, for a long time, leads to the discovery of cancerogenic effects. Therefore, the prediction of cancer risk in the investigated Wadi sediments detected by the radiological hazard parameter, namely, excess lifetime cancer risk (ELCR), which was calculated via the following Formula (8) [54]:

$$\text{ELCR(mSv/y)} = \text{AED}_{out} \times DL \times RF \tag{8}$$

where DL is the duration of life (DL = 70 years), Rf represents the cancer risk factor (RF = 0.05 Sv$^{-1}$), which was reported by the international commission of radiation protection (ICRP). Based on the descriptive statistics in Table 3, the mean value of ELCR (0.0005) is higher than the recommended value (0.00029), and the minimum and maximum values are $2.2 \times 10^{-4}$ and $8.5 \times 10^{-4}$, respectively.

## 5. Multivariate Statistical Analysis

The multivariate statistical analysis includes Pearson's correlation, a frequency distribution, Q–Q plot, principal component analysis (PCA), and hierarchical analysis (HCA). These analyses were conducted to view the association between the radionuclide activity concentrations and the radiological hazard indices.

### 5.1. Pearson's Correlation Analysis

Pearson's correlation was applied in the present study to identify the potency associations and linear relations between radionuclide activity concentrations and radiological

hazard indices from the Wadi sediments. Based on Pearson's correlation coefficient, the linear relation between the studied factors was classified into weak (0.00–0.19), moderate (0.2–0.39), strong (0.4–0.79), and very strong (0.8–1.00) correlations [55]. As can be reported in Table 5, the positive correlations are registered between all detected variables. Table 5 reveals that the radionuclides in the studied Wadi sediments are derived from natural sources, and their spatial distribution in the environment is not impacted by the other sources. The moderate correlations between radionuclide activity concentration and another in the studied Wadi sediments are observed. This indicates the sources of radionuclides in the investigated Wadi sediments differ slightly in nature. The correlations are strong and very strong regarding the relations among $^{238}$U, $^{232}$Th, $^{40}$K, and the radiological hazard parameters. $^{238}$U, $^{232}$Th, and $^{40}$K are the main contributors to the radiological hazards and the risks attributed to the emitted gamma radiation from radioactive chains in the Wadi sediments.

**Table 5.** Pearson's correlation matrix between radionuclide activity concentrations and radiological hazards parameters for the Wadi sediments.

|  | U-238 | Th-232 | K-40 | Raeq | Hin | Hex | Iγ | Dair | AEDout | AEDin | AGDE | ELCR |
|---|---|---|---|---|---|---|---|---|---|---|---|---|
| U-238 | 1 | 0.36 | 0.21 | 0.71 | 0.84 | 0.71 | 0.69 | 0.71 | 0.71 | 0.71 | 0.70 | 0.71 |
| Th-232 |  | 1 | 0.26 | 0.87 | 0.77 | 0.87 | 0.87 | 0.85 | 0.85 | 0.85 | 0.85 | 0.85 |
| K-40 |  |  | 1 | 0.52 | 0.45 | 0.52 | 0.56 | 0.56 | 0.56 | 0.56 | 0.58 | 0.56 |
| Raeq |  |  |  | 1 | 0.98 | 0.99 | 0.99 | 0.99 | 0.99 | 0.99 | 0.99 | 0.99 |
| Hin |  |  |  |  | 1 | 0.98 | 0.97 | 0.98 | 0.98 | 0.98 | 0.97 | 0.98 |
| Hex |  |  |  |  |  | 1 | 0.99 | 0.99 | 0.99 | 0.99 | 0.99 | 0.99 |
| Iγ |  |  |  |  |  |  | 1 | 0.99 | 0.99 | 0.99 | 0.99 | 0.99 |
| Dair |  |  |  |  |  |  |  | 1 | 1 | 1 | 0.99 | 1 |
| AEDout |  |  |  |  |  |  |  |  | 1 | 1 | 0.99 | 1 |
| AEDin |  |  |  |  |  |  |  |  |  | 1 | 0.99 | 1 |
| AGDE |  |  |  |  |  |  |  |  |  |  | 1 | 0.99 |
| ELCR |  |  |  |  |  |  |  |  |  |  |  | 1 |

### 5.2. Frequency Distribution and Q-Q Plot

The frequency distribution analysis and Q–Q plot are applied to the normality distribution of the $^{238}$U, $^{232}$Th, and $^{40}$K activity concentrations of the Wadi sediments and presented in Figure 6. Figure 6 indicates an asymmetrical distribution of the radionuclide dataset with positive and negative values of skewness. In the investigated Wadi sediments, the $^{238}$U and $^{232}$Th activity concentrations are positively skewed, while $^{40}$K is negatively skewed. However, the $^{238}$U and $^{232}$Th activity concentrations revealed the multimodality degree. This exposes that the Wadi sediments are enriched with different uranium and thorium minerals. The mineral analysis has confirmed the presence of various heavy minerals, such as uranothorite, monazite, zircon, allanite, and yettrocolombite. The Q–Q plot is another analysis that was used to confirm the normality distribution of the radionuclide dataset. The normality distribution was achieved when the observations were located on the right side of 45°. In the Q–Q plot, the observed data are presented on the x-axis, and the expected values for the normality distribution are shown on the y-axis. As can be observed in Figure 6, the non-normality for all radionuclides and the figures illustrate the a concave property where the low and high data are placed below the red line. This shows the deviation between low and high radionuclide activity concentrations deposited in the Wadi sediments from the surrounding granitoids.

### 5.3. Hierarchical Cluster Analysis

The correlation analysis data seemed complicated on account of the numerous parameters. Nevertheless, by utilizing a hierarchical cluster analysis (HCA), the relationships amid the radioactive parameters were adequately detected and exposed qualitatively. The HCA is a data categorization methodology that consists of a collection of multivariate

algorithms for determining real data groups. Items are clustered together in such a manner that related objects belong to the same category. Within hierarchical clustering, the data with the highest similarity are categorized first, followed by the subsequent most similar data. The process is repeated until all of the data are categorized. A dendrogram is built using the degrees of similarity at which the data combine. A similarity of 100% denotes that the clusters are separated by zero distance from corresponding sample measurements, whilst a similarity of 0% shows that the clustering areas are as dissimilar as the least similar region. Ward's approach was used to do a cluster analysis in this study. Ward's approach is a linkage mechanism for computing the Euclidean distance between radionuclide activity concentrations and radiological factors [56] (Figure 7). Five clusters are presented in the dendrogram of the investigated data. The first cluster is composed of $^{238}$U and $^{232}$Th, while the second cluster includes 232Th, Dair, and Raeq. The third cluster contains $^{232}$Th, Dair, and Raeq, and the fourth cluster Dair, Ra$_{eq}$, H$_{in}$, AGED, I$\gamma$, H$_{ex}$, and AED$_{in,}$ and the fifth cluster includes AGED, I$\gamma$, H$_{ex}$, AED$_{in}$, AED$_{out}$, ELCR, and $^{40}$K. The implementation of an HCA described the radioactivity of the Wadi sediments as associated with the activity of the radionuclide concentrations, particularly uranium and thorium. The HCA data are observed to match with Pearson's correlation.

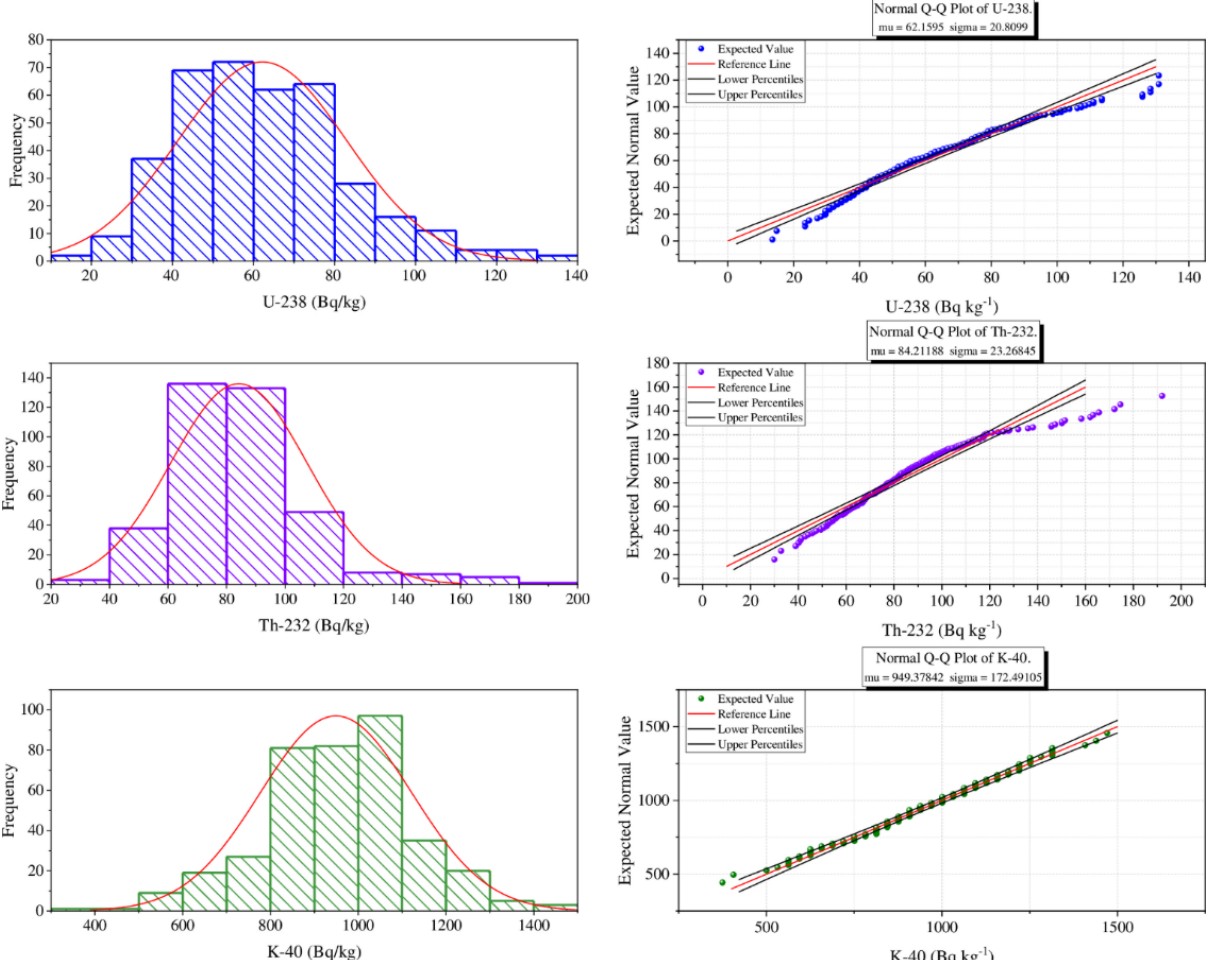

**Figure 6.** Frequency distribution analysis and Q–Q plot of $^{238}$U, $^{232}$Th, and $^{40}$K in the Wadi sediments in Wadi Rod Elsayalla.

### 5.4. Principal Component Analysis (PCA)

A factor analysis reduces a large number of variables to a few different types of components. The PCA results processing approach seeks to uncover any simple underlying structure within a multivariate results collection. For values greater than 0.7, loadings that demonstrate the relevance of the changeable for the elements are bolded. Characterizing

each item reveals a few high loadings and a lot of near-zero loadings, achieving the rotation's goals. According to Davis (1986), a maximizing variance involves raising the range of loadings, which drives to the extreme with negative or positive or near-zero loadings. In the current study, the PCA proceeded on the matrix correlation among various factors based on varimax rotations. The components PC1 and PC2 are listed in Table 6 and plotted in Figure 8. The $^{238}$U activity concentration possesses a high positive loading in the PC1 loading associated with all radiological parameters. The variance explained is 80.72%. This suggests the $^{232}$Th activity concentration as the main contribution of natural radioactivity in the sediments, followed by the $^{238}$U activity concentration at the area under study. On the other hand, 40 K has a weak negative loading in the PC2 load. The variance explained represents the percentage of 12.14. As can be observed, the loading variance is positive as explained by the fact that potassium that potassium does not contribute to the grade of the radiation exposure. The PC analysis shows that the total explained variance is 92.86%; thus, the radioactive data appears good [2].

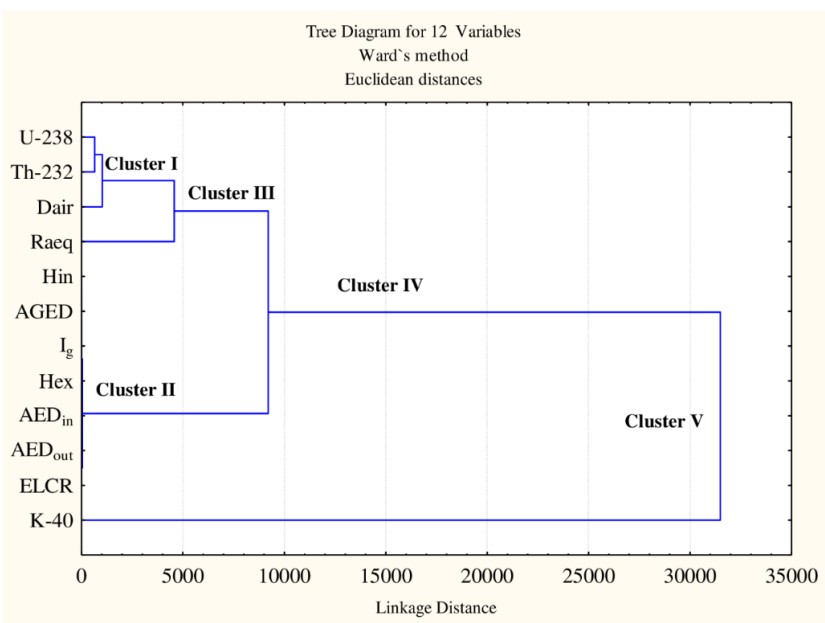

**Figure 7.** Dendrogram for the Wadi sediments in Wadi Rod Elsayallal.

**Table 6.** Principal component loadings and explained variances in the Wadi sediments.

| Radiological Parameters | Principal Component Analysis | | |
|---|---|---|---|
| | PC1 | PC2 | Communalities |
| U-238 | 0.62 | 0.74 | 0.38 |
| Th-232 | **0.90** | −0.28 | **0.81** |
| K-40 | 0.56 | −0.01 | 0.31 |
| Raeq | **0.99** | 0.12 | **0.98** |
| Hin | **0.94** | 0.32 | **0.89** |
| Hex | **0.99** | 0.12 | **0.98** |
| Ig | **0.99** | 0.10 | **0.99** |
| Dair | **0.99** | 0.14 | **0.98** |
| AEDout | **0.99** | 0.14 | **0.98** |
| AEDin | **0.99** | 0.14 | **0.98** |
| AGED | **0.99** | 0.12 | **0.98** |
| ELCR | **0.99** | 0.14 | **0.98** |
| Th-232/U-238 | 0.11 | −0.97 | 0.01 |
| Eigen value | 10.49 | 1.58 | |
| % Total variance | 80.72 | 12.14 | |
| Communalities % | 80.72 | 92.86 | |

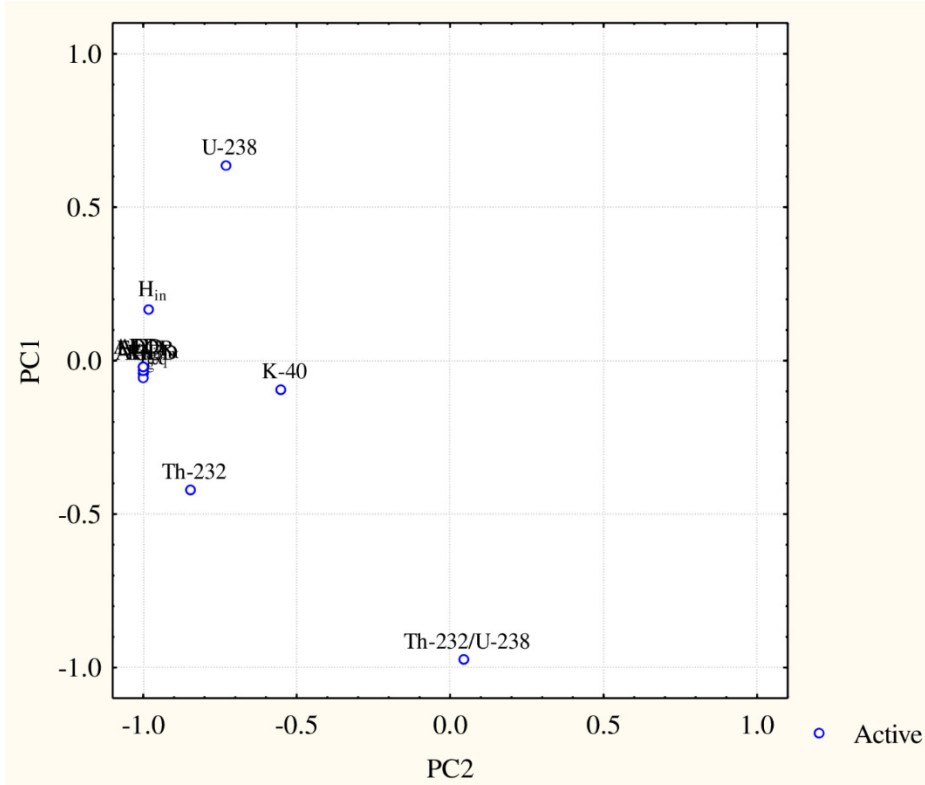

**Figure 8.** PCA in the Wadi sediments in Wadi Rod Elsayalla.

## 6. Conclusions

This study focused on two essential investigations. The first view is detecting heavy minerals from the surrounding granitoids that were derived from the Wadi sediments. The second view is the radioactive risk assessment for the human practicing daily activity in the studied area (Wadi Rod Elsayalla). The mineralized samples were collected from the highest radioactive zones from syenogranite and quartz syenite. These samples were subjected to disintegration (crushing and grinding). The study illustrated that the examined mineralized syenogranite and quartz syenite were enriched in heavy radioactive minerals, such as uranothorite as well as monazite, zircon, yttrocolumbite, and allanite. The activity concentration mean of $^{238}$U, $^{232}$Th, and $^{40}$K was 62.2 ± 20.8, 84.2 ± 23.3, and 949.4 ± 172.5 Bq kg$^{-1}$, respectively, and exceeded the recommended limit. The annual effective dose (AED) results displayed the exposure to gamma radiation, either outdoors −0.1 mSv y$^{-1}$ or indoors 0.6 mSv y$^{-1}$, and was higher than the recommended value. The annual gonadal dose equivalent (AGDE- 0.8 mSv y$^{-1}$) and excess lifetime cancer (ELCR- 0.0005) mean values exceeded the permissible value. The statistical analysis confirmed the radiological hazards were contributed by the radionuclides $^{238}$U, $^{232}$Th, and $^{40}$K. The risk was assigned to the gamma radiation emitted from the radioactive decay chains. Finally, the present study revealed the investigated Wadi sediments could not be applied in various building material and infrastructures applications. However, the radioactive minerals in the studied granitoid were responsible for higher natural radionuclides in the studied sediments in the Wadi Rod Elsayalla area.

**Supplementary Materials:** The following are available online at https://www.mdpi.com/article/10.3390/app112411884/s1, Table S1: the concentrations of radionuclides $^{238}$U, $^{232}$Th, and $^{40}$K and the radiological hazard indices.

**Author Contributions:** Conceptualization, A.E.A.G., K.A. and M.Y.H.; methodology, A.E.A.G. and H.E.; software, A.E.A.G., M.U.K. and M.Y.H.; validation, M.I.S. and D.A.B.; formal analysis, A.E.A.G., M.U.K. and M.Y.H.; investigation, M.I.S. and D.A.B.; resources, H.E. and B.H.E.; data curation, K.A.

and H.E.; writing—original draft preparation, A.E.A.G., M.I.S., M.U.K. and M.Y.H.; writing—review and editing, A.E.A.G., M.U.K. and M.Y.H.; visualization, M.I.S.; supervision, H.E. and D.A.B.; project administration, H.O. and B.H.E.; funding acquisition, H.O. and B.H.E. All authors have read and agreed to the published version of the manuscript.

**Funding:** The authors acknowledge the support of Taif University Researchers Supporting Project number (TURSP-2020/127), Taif University, Taif, Saudi Arabia.

**Institutional Review Board Statement:** Not applicable.

**Informed Consent Statement:** Not applicable.

**Data Availability Statement:** Not applicable.

**Acknowledgments:** The authors acknowledge the support of Taif University Researchers Supporting Project number (TURSP-2020/127), Taif University, Taif, Saudi Arabia.

**Conflicts of Interest:** The authors declare no conflict of interest.

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
