# Peer review of "Radiological Investigation on Sediments: A Case Study of Wadi Rod Elsayalla the Southeastern Desert of Egypt"

_applsci, doi:10.3390/app112411884_

Round 1

Reviewer 1 Report

The publication requires significant improvements. In particular, the research methodology should be improved. It should be clarified how many samples were tested, whether radiometric tests were carried out in the field or on samples, if so, of what density and fraction. All comments are included in the publication. 

Author Response

Response to Reviewer

Ref: Revision of the manuscript applsci-1477931

Title: “Radiological investigation on sediments: a case study of  Wadi Rod Elsayalla, the South Eastern Desert of Egypt”

Authors: Ahmed E. Abdel Gawad, Khaled Ali, Hassan Eliwa, M. I. Sayyed, Mayeen U. Khandaker, David A. Bradley, Hamid Osman, Basem H. Elesawy, Mohamed Y. Hanfi.

Please find attached the submission of the carefully revised version of the manuscript in Ref., following the comments and modification of the Reviewer.

Below, the detailed list of the changes made in response to the Reviewer’s major comments (in italics), outlining every change made point by point, is provided. The changes are marked in the manuscript text.

Reviewers' comments:

Questions

Answer

Reviewer 1

You have described the heavy minerals found in granites, what the heavy minerals in sediment? Explain this

Wadi Rod Elsayalla sediments were derived from the surrounding basement rock in the area under investigation.

So, the heavy minerals could be related to the host basement rocks from which the Wadi sediments were derived.

granitoids

added

The mean activitiy of 238U, 232Th and 40K concentrations are 62.2±20.8, 84.2±23.3 and 949.4± 172.5 Bq kg-1 respectively […] But where in the sediments?

The mean activity of 238U, 232Th and 40K concentrations are 62.2±20.8, 84.2±23.3 and 949.4± 172.5 Bq kg-1, respectively for the investigated Wadi sediments,

Change the font size.

done

Remove the full stop

done

space

done

space

done

Point 2. Geological setting

Next 3 Materials and Methods

Change it.

done

Give the geographical background of The Wadi Rod Elsayalla

Wadi Rod Elsayalla area is located in the South Eastern Desert of Egypt. It can be reached through asphalic rod about 120 km eastward of Aswan city and occurs between latitude 23o 50′ 47′′ and 23o 53′ 28′′ N and longitude 34o 20′ 52′′ and 34o 23′ 44′′ E (Figure 1a, b).

Change the font size.

done

What is the mineral composition of metavolcanic rocks?

Meta-basaltic rocks are composed essentially of plagioclase laths and pyroxene set in a fine-grained groundmass. Meta-andesitic rocks are of the porphyritic type, are composed of plagioclase hornblende, biotite, quartz and iron oxides. Meta-dacite rocks are composed mainly of plagioclase, K-feldspar, quartz and fine biotite flakes set in a very fine-grained groundmass. Meta-rhyolite rocks are composed of quartz, K-feldspars and plagioclase phenocrysts embedded in the fine groundmass.

What is the age of the all rocks?

The metavolcanic have an age dated between 860 and 825 Ma (El-Shazly et al., 1973).

The granitoids rocks in the Eastern Desert of Egypt vary in their composition from syenogranite, monzogranite to alkali feldspar granite and they are dated between 610 and 550 Ma during the Late-African Orogeny (Stern and Hedge, 1985; Ali 2014; Eliwa et al., 2014; Skublov et al., 2021; Abdel Gawad et al., 2021).

The younger granite intrusions and their related pegmatites are dated 610 ± 20 and 594 ± 12 Ma (Moghazi et al., 2004).

If you have a photo of the source rocks show it.

added

What is the age of the sediments?

The Wadi Rod Elsayalla are composed essentially of recent sediments derived from the surrounding granitoids

This map is unclear. Change the quality.

Where on the map are the sample locations?

A - Physiographic map and next B - Detailed geological [...]

First there must be the materials.

Have done

3.1 point

done

GR or GS - Gamma Spectrometer

GS-256 Gamma Spectrometer

GS

added

That's not true. What the authors understand by "background". Each type of rock has its own "background". Perhaps the authors should consider the need to measure the cosmic background. Such a measurement is made in a large body of water, away from the shores, so as to exclude radiation from the rocks. There is an abundant literature on this subject. Moreover, the article does not provide any information on the altitude above sea level of the described research area. At high altitudes (over 1000 m above sea level) it is necessary to make a correction for the cosmic background.

Thanks for your interpretation. The authors are agreed with your opinion. Each rock has its own background. Thus, the authors have removed the paragraph from the text. Also, the cosmic radiation in the investigated area didn’t detect. In future work, the authors will detect the doses are associated with exposure to cosmic radiation.

How many sediment samples were taken?

How many samples of heavy minerals did you obtain?

380 stations for ground gamma-ray spectrometric measurements have been taken at Wadi Rod Elsayalla sediments, as mentioned in Table S1

Ten samples were selected from the mineralized zones from syenogranite and quartz syenite in order to investigate the radio-elements –bearing minerals

There is no description of the sampling method

Ten mineralized samples were collected from the highest radioactive zones from syenogranite (5) and quartz syenite (5) in order to investigate the heavy minerals, especially the radioactive ones (Figure 1b). These samples were subjected to disintegration (crushing and grinding). Samples were washed using water; slimes were removed then samples were dried. The dried fraction is exposed to be sieved into the size -0.5 to +0.063 mm. The systematic mineral separation technique includes a water table and heavy liquid (Bromoform, 2.8 g/cm3 for specific gravity) separating the light and heavy fractions. Also, polished sections were selected from granitoids for mineralogical investigation.

Do you study heavy minerals in rock or in sediment? The title talks about sediments and not rocks. Explain.

As mentioned before

Wadi Rod Elsayalla sediments were derived from the surrounding basement rock in the area under investigation.

So, the heavy minerals could be related to the host basement rocks from which the Wadi sediments were derived.

specific gravity

2.8 g/cm3 for specific gravity

Next methods of separation!

3.3. Methods of separation

This in next point. Methods!

3.3. Methods of separation

What was used to calibrate the equipment and minerals?

EDX calibration standard has been specifically designed to help calibrate any EDX system.

Standards block contains:

- Mn (Manganese for EDS detector resolution test).

- Cr (Chromium for light element sensitivity tests).

- C (Carbon for light element sensitivity tests).

- Ni (Nickel for sensitivity tests).

- PTFE (Polytetrafluoroethylene for Flourine light element tests)

- Co (Cobalt for Quant optimization).

- Almandine Garnet (for Quant tests).

- Faraday Cup (Hole size 150um. For accurate probe/specimen current measurement. Having a stable probe current is vital for achieving repeatable analysis results).

- Silicon magnification grid (For calibration of magnification levels within an SEM. It is marked with clearly visible squares of periodicity 10 µm. The dividing lines are about 1.9 µm in width and are formed by electron beam lithography).

How many heavy minerals are in the samples? The amount of heavy minerals (containing U and Th) determines the radiation of the whole sample.

A chemical composition study of the bulk sample should be done.

This work aims to investigate heavy minerals that are responsible for radioactivity

Heavy minerals are uranothorite, monazite-(Ce), zircon, yttrocolumbite and allanite

The whole-rock geochemical data of granitoids have been published before, as reported by (Eliwa et al., 2018).

This citation is mentioned in section references.

What is the mineral composition of the whole samples?

Syenogranite is medium to coarse-grained, reddish to pale pink in colors. It is highly weathered, jointed, strongly altered and exfoliated. Syenogranite is composed principally of K-feldspar, quartz, biotite, plagioclase and iron oxides. They contain xenoliths up to 1m of subangular metatuffs along their outer periphery. Alkali feldspar granite is coarse-grained, whitish to pale white colors, and highly weathered. Quartz syenite is medium to coarse grained, dark grey to pale greenish grey, pale pink in colors and moderate to high relief. It is highly weathered, exfoliated, and composed of K-feldspars, quartz, biotite, riebeckite, arfvedsonite and a very subordinate amount of plagioclase.

Give the chemical formula

uranothorite, monazite, zircon and yttrocolumbite

Uranothorite (Th, U) SiO4

Monazite-(Ce) (Ce, La, Pr, Nd, Th, Y)PO4

Zircon (ZrSiO4)

The yttrocolumbite mineral is one of the columbite groups that comprises a general formula AB2O6, A = Fe, Mn Y, REE, Ca, U, and Th, and B = Nb and Ta.

Allanite-(Ce) (Ca; Ce)2(Al; Fe2+; Fe3+)3(SiO4)(Si2O7)O(OH)

Monazite has no of Th and U inclusions ?

ThO2 could be reached up to 8.25 wt%, and 11.69 wt%, and UO2 contents reached 0.57 and 4.83 wt% in monazite (Ali et al., 2021; Abdel Gawad et al., 2021).

index

Zr2 and Zr5 (Table 1, Figure 3).

Give the chemical formula

Allanite

Allanite-(Ce) (Ca; Ce)2(Al; Fe2+; Fe3+)3(SiO4)(Si2O7)O(OH)

This is thin section. Where is the description of how many analyese were done?

as mentioned in (Tables 1 and 2)

It was in the introduction

Deleted text

It is not clear whether the spectrometric measurements were performed directly in the field or whether the radioactivity of 380 samples collected in the field was tested in the laboratory.

The spectrometric measurements were performed directly in the field for the investigated Wadi sediments.

380 stations for ground gamma-ray spectrometric measurements have been taken at Wadi Rod Elsayalla sediments as mentioned in Table S1

This is not on the map.

Deleted text

It is difficult to argue with this truth, which is obvious to geochemists. But the uranium concentration process and the calcrete urine deposit are also known. Did the authors not observe this phenomenon in the studied area?

Calcrete uranium deposits have not occurred in the studied area.

Greek

Greece

The samples are not from sediments but from rocks. Quote below:

The mineralized samples were collected from the highest radioactive zones from syenogranite and quartz syenite. These samples were subjected to disintegration (crushing and grinding).

Yes,

The mineralized samples were collected from the highest radioactive zones from syenogranite and quartz syenite. These samples were subjected to disintegration (crushing and grinding).

It is obvious that the Wadi Rod Elsayalla sediments were derived from the surrounding granitoids

Quantitative and qualitative data are lacking.

Chemical composition of samples is missing.

380 stations for ground gamma-ray spectrometric measurements have been taken at Wadi Rod Elsayalla sediments, as mentioned in Table S1

The whole-rock geochemical data of granitoids have been published before, as reported by (Eliwa et al., 2018).

This citation is mentioned in section references.

We thank a lot the Reviewer for the useful and valuable comments that have helped to improve the manuscript.

Hoping that all the careful review is sufficient for the direct acceptance of the manuscript, thank you for your time and consideration.

Best wishes,

Mohamed. Y. M. Hanfi

on behalf of all co-authors

Reviewer 2 Report

Dear Author,

The paper is well-written and looks fine, however some small correction requires pirror to acceptance. please see the attached file.

Regards,

Author Response

Ref: Revision of the manuscript applsci-1477931

Title: “Radiological investigation on sediments: a case study of  Wadi Rod Elsayalla, the South Eastern Desert of Egypt”

Authors: Ahmed E. Abdel Gawad, Khaled Ali, Hassan Eliwa, M. I. Sayyed, Mayeen U. Khandaker, David A. Bradley, Hamid Osman, Basem H. Elesawy, Mohamed Y. Hanfi.

Please find attached the submission of the carefully revised version of the manuscript in Ref., following the comments and modification of the Reviewer.

Reviewer 2

It is too long, I would snuggest something like:

Radiological investigation on sediments: case study of  Wadi Rod Elsayalla, the South Eastern Desert of Egypt

Radiological investigation on sediments: a case study of  Wadi Rod Elsayalla, the South Eastern Desert of Egypt

why do you think such estimation would be applied for this study? why do you calculate internal hazard indices, while it belongs to indoor radiation assessment from building materials.

The authors believe the public around the surrounding area can be used the sediments in the building materials and the different infrastructures applications. Therefore, the internal dose can be predicted from the radon gas and its decay products are released from the sediments.

Again why this calculation was important in your case?

The gamma level index is associated with the natural radionuclides in building materials.

The authors also believed the exposure to gamma radiation from the sediments are applied in the building materials. Thus, the public can be transferred the sediments to apply in various activities such as infrastructures fields.

Below, the detailed list of the changes made in response to the Reviewer’s major comments (in italics), outlining every change made point by point, is provided. The changes are marked in the manuscript text.

We thank a lot the Reviewer for the useful and valuable comments that have helped to improve the manuscript.

Hoping that all the careful review is sufficient for the direct acceptance of the manuscript, thank you for your time and consideration.

Best wishes,

Mohamed. Y. M. Hanfi

on behalf of all co-authors

Round 2

Reviewer 1 Report

Dear Authors, 
the changes you have made have improved the quality of your manuscript. I have no comments. The manuscript can be published in present form.

Author Response

Thank you very much for your comments and recommendations. The authors hope that the manuscript will be useful to researchers. 

Best Wishes 

Mohamed Hanfi